# Effects of 1-MCP Treatment on Physiology and Storage Quality of Root Mustard at Ambient Temperature

**DOI:** 10.3390/foods11192978

**Published:** 2022-09-23

**Authors:** Wenyan Lin, Yaping Liu, Jianbing Di, Gang Ren, Wei Wang, Weichun He, Yu Wang

**Affiliations:** College of Food Science and Engineering, Shanxi Agricultural University, Jinzhong 030801, China

**Keywords:** 1-MCP, root mustard, physiology, storage quality

## Abstract

Root mustard is plentiful in vitamins and minerals but shrivels and molds easily. In this study, freshly harvested root mustard was fumigated with various concentrations of 1-Methycyclopropene(1-MCP) (1 µL L^−1^, 1.5 µL L^−1^, and 2.0 µL L^−1^) for 24 h and stored at ambient temperature (17 ± 1 °C) for 35 d. Our data showed that 1-MCP fumigation had a striking preservation effect on maintaining weight loss, fruit firmness, lignin, Vc content, and moisture content, inhibiting respiratory intensity and ethylene release rate, as well as decreasing cell permeability and malondialdehyde (MDA) accumulation and maintaining cell membrane integrity of root mustard. In addition, lipoxygenase (LOX), pyruvate dehydrogenase (PDH), and polyphenol oxidase (PPO) activities were significantly reduced throughout the storage period. In contrast, the activities of succinate dehydrogenase (SDH), superoxide dismutase (SOD), ascorbate peroxidase (APX), phenylalanine deaminase (PAL), and peroxidase (POD) remained at high levels. Results showed that 1-MCP treatments were effective in maintaining the quality of root mustard, and the preservation effect of 1.0 μL·L^−1^ 1-MCP was better than other concentrations of 1-MCP. This study could serve as a theoretical reference for root mustard preservation.

## 1. Introduction

Root mustard (*Brassica juncea var. megarrhiza Tsen et Lee*), also known as kohlrabi and turnip, is a genus of *Brassica* in the cruciferous family [1,2], which is widely distributed in southwest China and Yangtze River basins [3,4]. Root mustard has a high nutritional value and is rich in various essential amino acids for the human body. Root mustard has diverse effects, including antibacterial, hypolipidemic, antitumor, anti-aging, and refreshing effects [5,6]. However, after harvesting, root mustard is highly respiratory and prone to water loss, shriveling, nutrient loss, and mold, thus resulting in substantial losses. Therefore, root mustard urgently needs green and efficient storage technology to maintain its good quality and commodity value.

At present, various post-harvest preservation methods, including low temperature [7,8], biochemical treatment [9,10,11], and physical treatment [12,13], have been used to delay senescence, extend shelf life and preserve the quality of fruits and vegetables [14]. However, these preservation methods present several shortcomings and limitations, including short storage time at low temperature, high physical treatment costs, and some chemical preservation may produce potential hazards and pollution problems. To alter the above phenomena, 1-MCP has gradually attracted attention in recent years. 1-MCP is an ethylene receptor inhibitor cyclopropene compound, which can irreversibly act on ethylene receptors, thereby preventing regular ethylene binding and restraining a range of physiological and biochemical responses to fruit ripening [15,16]. Compared to the conventional fresh-keeping agent, 1-MCP has the advantages of non-toxicity, good chemical stability, easy synthesis, and low use concentration [17]. Some research has demonstrated that 1-MCP takes an essential role in the post-harvest physiological processes of fruits and vegetables. For instance, Cefola [18] found that using 1-MCP markedly extended the shelf life of fresh-cut broccoli, reduced post-harvest spoilage, and slowed the degradation of chlorophyll. Studies on sweet potato [19,20] have shown that 1-MCP reduces germination, increases soluble solids levels, delays the degradation of ascorbic acid content, and inhibits the increase in MDA content, while maintaining a good antioxidant capacity. Yu et al. [21] found that 1-MCP treatment inhibited the rise in mitochondrial ROS content of courgette, maintained the activity of antioxidant enzymes SOD and APX, and energy metabolizing enzymes SDH, CCO, H^+^-ATPase, and Ca^2+^-ATPase, and maintained normal energy metabolism in fruit tissues. 1-MCP has been proven to postpone maturation in a series of fruits and vegetables, which include apples [22,23], pears [24,25], peaches [26,27], and ginger [28]. Till now, root mustard research has mainly focused on mustard fermentation [29,30], strain isolation [4], etc., with less research on its preservation, while storage methods have mainly focused on traditional methods [31] (burial method, cellar method, plastic film bagging storage method, stacking storage method), low temperature storage and air conditioning storage. Sun [32] found that low temperature (4 °C) inhibited the decline in the sensory quality of small mustard, and a low-temperature refrigeration method was found to be more effective in maintaining the freshness, nutritional value, and original flavor of the mustard, but its storage time was shorter. Luo et al. [33] showed that simple gas preservation can effectively reduce the weight loss rate of stem mustard, reduce the loss of nutrients, and maintain better color and freshness. So far, the effect of 1-MCP on the storage quality of root mustard has rarely been studied. Studies have shown that the preservation effect of 1-MCP is related to the treatment temperature. Serek et al. [34] found that the preservation effect of high temperature treatment is better than that of low temperature in a certain temperature range. Ku et al. [35] found that high temperature conditions are more conducive to the binding of 1-MCP to ethylene sites. Sisler et al. [16] found that the reason for the significant decrease in 1-MCP effect under low-temperature conditions may be related to the change of receptor protein conformation on the membrane at low temperatures. In this experiment, 1-MCP at concentrations of 1.0 µL L^−1^, 1.5 µL L^−1^, and 2.0 µL L^−1^ was selected to treat root mustard based on previous studies on other root vegetables. The effects of 1-MCP treatment with different concentrations on the postharvest storage properties of root mustard at ambient temperature were studied, which provided a reference for the storage technology and preservation of mustard.

## 2. Materials and Methods

### 2.1. Materials

Root mustard, a variety of bald mustard, was harvested in Taigu District, Jinzhong City, Shanxi Province, and the stems and leaves of root mustard were removed. Selection of samples without mechanical damage, pests, and diseases, uniform size to be used; 0.087 g, 0.1305 g, and 0.1740 g 1-MCP powders were used to configure 1-MCP with concentrations of 1.0 μL·L^−1^,1.5 μL·L^−1^, and 2.0 μL·L^−1^, respectively.

### 2.2. Sample Handling

The root mustard was randomly divided into four groups of 85 and placed in a 60 L plastic barrel. The root mustard was sealed fumigation treatment for 24 h by using different concentrations (0 μL·L^−1^, 1 μL·L^−1^, 1.5 μL·L^−1^, 2.0 μL·L^−1^) of 1-MCP (Aladdin Bio-Chem Technology Co., Ltd. Shanghai, China.), and the cover was opened for ventilation for an hour. Then, the root mustard was packed in 0.03 mm perforated polyethylene plastic bag and stored at ambient temperature (17 ± 1 °C) with a relative humidity of 65% to 75% for 35 d. The relevant indexes were measured in 7-day intervals, and the experiment was repeated three times.

### 2.3. Storage Effect and Quality

#### 2.3.1. Measurement of Weight Loss and Hardness

Weight loss rate was ascertained by the weighing approach, and initial mass was fixed for each group, recorded as m/g, and the mass measured every 7 days was recorded as m_1_/g. The weight loss (%) is based on the equation below (1).
(1)$$\mathrm{weight}\;\mathrm{loss}\;\mathrm{rate}\;(\%)=\frac{m-m_{1}}{m}\;\times \;100$$

Hardness measurements were conducted at ambient temperature using a food texture analyzer (FTA) (TMS-PRO, American TFC Company). The central part of the root mustard (1 cm × 2 cm × 1 cm) was taken and placed on a plate, then a cylindrical plunger with a diameter of 3 mm was pushed into the sample at a speed of 100 mm·min^−1^ [36,37]; 9 points were tested for each treatment. Finally, the average value was taken.

#### 2.3.2. Measurement of Respiratory Intensity and Ethylene Content

Root mustard was transferred to a 2.5 L plastisol box, allowing carbon dioxide and oxygen to accumulate in the box at 17 °C. After 2 h, gas analyzer was measured using a carbon dioxide analyzer (F-940, Felix Instruments Inc., Camas, WA, USA), calculated according to Formula (2).
(2)$$\mathrm{Respiratory}\;\mathrm{intensity}\;(\mathrm{mg}\cdot {\mathrm{kg}}^{-1}\cdot h^{-1})=(\frac{M}{22.4}\cdot N\cdot \frac{273}{273+T})\cdot V\cdot (m\cdot h{)}^{-1}$$
m represents sample mass, M represents the relative molecular mass of gas, T represents ambient temperature, N represents CO_2_ concentration, V represents the volume of the container used for determination, and h represents measurement time.

Two root mustards were transferred to a 2.5-litre plastic box with a pinhole-sized hole on the lid for subsequent determination and sealing, allowing ethylene to accumulate in the box at 17 °C. After 24 h, gas analyzer (F-940, Felix Instruments Inc., Camas, WA, USA) was connected to the injection head to measure the ethylene content.

#### 2.3.3. Measurement of Cell Membrane Permeability and MDA Content

Cell membrane permeability refers to the approach of Wei et al. [38] with some modifications. The sample was punched into 10 holes with a puncher with a diameter of 10 mm, washed three times with distilled water, and immersed in 25 mL distilled water in a 50 mL test tube for 30 min to determine the initial conductivity (C_1_). The solution was boiled for 30 min, allowed to cool, and then the maximum conductivity (C) was measured. The experiment was repeated 3 times and the cell membrane permeability was represented by relative conductivity. The relative conductivity is calculated according to Formula (3).
(3)$$\mathrm{Relative}\;\mathrm{conductivity}\;(\%)=\frac{C_{1}}{C}\;\times \;100\%$$

Malondialdehyde (MDA) content was assayed as described by Zhang & Liu [39] and was expressed as μmol·g^−1^. The sample (1 g) was ground to homogenate with 5% trichloroacetic acid (2 mL) and quartz sand, and then ground again after adding trichloroacetic acid (8 mL). The homogenate was centrifuged at 4000 r·min^−1^ for 10 min, and the supernatant was collected. Add 2 ml 0.6% thiobarbituric acid solution to the supernatant and mix well. The mixture was placed in boiling water for 10 min, and then, after cooling, centrifuged at 3000 r·min^−1^ for 15 min. The absorbance values at 532 nm, 600 nm, and 450 nm were determined with 0.6% thiobarbituric acid solution as blank. The experiment was repeated three times. MDA content is calculated according to Equation (4).
(4)$$\mathrm{MDA}\;\mathrm{content}=\frac{\left({\mathrm{OD}}_{532}-{\mathrm{OD}}_{600}\right)\times V_{1}}{1.55\times 0.1\times V_{T}\times W}$$

V_1_ is the volume of sample taken for the determination/mL; V_T_ is the total volume of the extract/mL; W is the mass of the sample/g; 1.55 × 10^−1^ is the micromolar extinction coefficient of MDA.

#### 2.3.4. Measurement of Vit C Content and Lignin Content

Vit C content was determined by titration of 2,6-dichlorophenol indigo [40] and denoted by mg·100 g^−1^. Samples of root mustard were ground into a homogenate in 2% oxalic acid solution. The final volume was 100 mL. After filtering, the 10 mL filtrate was titrated with the calibrated 2.6-dichloro indophenol sodium solution. When the color of the solution changed to pink, the dye usage was noted. Vit C content was calculated according to Equation (5).
(5)$$W=\frac{\left(V-V_{1}\right)}{B}\times \frac{A\times b}{a}\times 100$$

W means the 100 g sample contains mg of ascorbic acid; V is the number of ml of dye used to titrate the sample; V_1_ is the milligrams of dye used in blank titration; A is 1 mL dye solution equivalent to milligrams of ascorbic acid; B is the number of milliliters of the sample solution taken during titration; b is the total milliliters of sample solution after dilution; a is the number of grams of sample.

The lignin content was assayed according to the approach of Zhi et al. [41] with some modifications. Root mustard (0.1 g) was extracted with 95% ethanol (10 mL) for 20 min. Next, centrifuged at 8000 r·min^−1^ for 20 min, repeated twice, and the supernatant was poured out. The ethanol: n-hexane = 1:2 (V:V) was extracted for 20 min, then centrifuged at 8000 r·min^−1^ for 20 min and repeated twice. Afterward, the precipitate was dried at ventilation for 20 h, and 25% bromoacetyl glacial acetic acid (3 mL), 2 mol·L^−1^ sodium hydroxide (0.9 mL), 5 mL acetic acid and 7.5 mol·L^−1^ hydroxylamine hydrochloride (0.1 mL) were added into the dry matter. Finally, acetic acid was fixed to 15 mL, and the absorbance values were measured at 280 nm. The lignin content was calculated according to Equation (6).
lignin content (U·g^−1^) = (A_280_ × V_r_) × (0.01 W × V_s_)^−1^
(6)

Vr represents the total volume of the extract (mL), W represents sample quality (g), and Vs represents the liquid volume of enzyme used for determination (mL).

#### 2.3.5. Measurement of Moisture Content

Root mustard (1 g~3 g) was placed on the sample plate, the sample room was closed, and the moisture content in the sample was determined according to the instrument operation procedure.

### 2.4. Determination of Enzyme Activity

#### 2.4.1. LOX, SOD, PAL, APX, PDH and SDH Activity

Lipoxygenase (LOX) activity was determined by reference to the method of Ke et al. [42] with some modifications. Substrate solution: Linoleic acid (0.27 mL) and Tween 20 (0.25 mL) were added to sodium hydroxide (1.0 mol·L^−1^, 5.0 mL) and all compounds were mixed equally. The pH of the mixture was adjusted to 9.0 with hydrochloric acid, and the volume was adjusted to 500 mL with borate buffer (pH = 9.0). Crude enzyme solution: Samples (0.5 g), 20 mL phosphate buffer (pH = 7.0) and 20 mL polyvinylpyrrolidone solution were added to the mortar, respectively. After fully ground, the grinding solution was centrifuged at 4 °C, 12,000 r·min^−1^ for 30 min, and the middle clear liquid was filtered by using a 0.25 μm filter membrane. At room temperature (25 °C), the substrate solution (0.9 mL), borate buffer (0.2 mol·L^−1^, 2.0 mL and pH = 9.0) and crude enzyme solution (0.1 mL) were added to the 1 cm quartz cuvette. Enzyme activity was determined spectrophotometrically by measuring the increase in absorbance at 234 nm over a 1 min period.

Superoxide dismutase (SOD), phenylalanine deaminase (PAL), ascorbate peroxidase (APX), pyruvate dehydrogenase (PDH), and succinate dehydrogenase (SDH) activity were measured according to the instructions of the Solarbio (Solarbio Science & Technology Co., Ltd. Beijing, China) kit.

#### 2.4.2. PPO and POD Activity

Polyphenol oxidase (PPO) and peroxidase (POD) activities were measured by Cao [40]. Enzyme solution preparation: 2 g sample was weighed and put into a mortar, phosphate-buffered solution (8 mL, pH 6.4) was added and homogenized in an ice bath, then centrifuged at 12,000 r·min^−1^ at 4 °C for 30 min. Determination of PPO activity: supernatant (0.6 mL) and buffer (0.3 mL) were added to the test tube, and balanced in a water bath at 30 °C for 5 min, then 0.05% catechol (3 mL) was added, and the OD_420_ value was measured immediately after shaking. Record once every 30 min for a total of 5 min, an increment of 0.01 in OD_420_ per minute was regarded as one unit of enzyme activity. Determination of POD activity: supernatant (0.5 mL) and 0.1% guaiacol (2 mL) were added to the test tube, and test tube was equilibrated at 30 °C water bath for 5 min, then 0. 18% H_2_O_2_ (1 mL) was added, and the OD_470_ value was measured immediately after shaking. We recorded once every 30 min for a total of 5 min, and an increment of 0.01 in OD_470_ per minute was regarded as one unit of enzyme activity.

### 2.5. Statistical Analysis

Statistical analyses were performed with SPSS Statistics 23 (IBM SPSS Statistics, Armonk, NY, USA). Significant differences were ascertained using Duncan’s multiple range tests at the level of *p* < 0.05.

## 3. Results

### 3.1. Storage Quality of Root Mustard

#### 3.1.1. Weight Loss Rate and Hardness

Weight loss rate and hardness are essential parameters that reflect the texture of root mustard. It can be seen from Figure 1 that 1-MCP fumigation significantly decreased the weight loss rate of root mustard compared to the control (*p* < 0.05). On the 35th day of storage, the weight loss rate of the control group and 1.0 μL·L^−1^, 1.5 μL·L^−1^, and 2.0 μL·L^−1^ treatment groups were 4.58%, 1.79%, 1.81%, and 1.89%, respectively. Throughout storage, the control root mustard showed an overall decreasing trend in hardness due to nutrient depletion (Figure 1B), whereas 1-MCP-treated root mustard first dropped and then rose in hardness, reaching the peak at 21 days and then gradually decreasing. During storage, root mustard hardness was better in the 1-MCP fumigation groups than in the control group, and the hardness of root mustard treated with 1.0 μL·L^−1^ 1-MCP was the best. Therefore, 1-MCP fumigation could better inhibit the reduction of postharvest root mustard hardness and was most effective at a concentration of 1.0 μL·L^−1^.

#### 3.1.2. Respiratory Intensity and Ethylene Release Rate

Measurements of the respiratory intensity and ethylene release rate are presented in Figure 2A,B, respectively. During the whole storage period, the respiration intensity of root mustard showed a fluctuating state. On the 7th–35th day of storage, the respiratory peaks of all 1-MCP fumigation groups were lower than that of the control group, and 1.0 μL·L^−1^ was the lowest (Figure 2A). We monitored the production of ethylene in control and 1-MCP-treated root mustard at various times (Figure 2B). Compared with the control group, 1-MCP fumigation groups both had an inhibitory effect on ethylene release during the storage of root mustard. On the 35th day of storage, ethylene release from root mustard was 0.457 mg·kg^−1^·h^−1^, 0.198 mg·kg^−1^·h^−1^, 0.224 mg·kg^−1^·h^−1^, and 0.225 mg·kg^−1^·h^−1^ for each group, with 1.0 μL·L^−1^ 1-MCP being released at the lowest level, respectively. Thus, respiration of root mustard was inhibited, and ethylene release was reduced by fumigation with concentrations of 1.0 μL·L^−1^ 1-MCP.

#### 3.1.3. Cell Permeability and MDA Content

The cell permeability of root mustard first increases, followed by a decrease, reaching a peak at 28 days. The lowest cell permeability of root mustard was observed in the 1.0 μL·L^−1^ and 1.5 μL·L^−1^ groups with peaks of 9.21% and 9.32%, which was significantly lower than the control (18.18%) (*p* < 0.05) (Figure 3A). As shown in Figure 3B, the MDA content of root mustard in control displayed an increasing tendency. In contrast, the MDA content in root mustard treated with 1-MCP showed a downward trend, which had been maintained at a low level. In addition, root mustard MDA content was significantly more in control than in the 1-MCP fumigated throughout the storage period (*p* < 0.05). In summary, this indicated that 1-MCP fumigation could alleviate the cell membrane damage of root mustard and prolong the storage time of root mustard.

#### 3.1.4. The Vit C Content and Lignin Content

Vit C is a hexolactone compound synthesized in plants. It is not only an essential substance for maintaining human health, but also has special functions for plants in maintaining the redox balance of cell photosynthesis and metabolism [43]. The Vit C content of root mustard showed a downward trend, and the control root mustard had a lower Vit C content than other groups. No significant differences were detected between pre-storage treatment groups, but as time passed, the advantage of concentration 1.0 μL·L^−1^ gradually emerged (Figure 4A). The lignin content in control displayed an overall uptrend with an extensive variation range (Figure 4B). In contrast, the lignin content of 1-MCP treatment changed little and tended to be gentle. There was no significant difference between treatments throughout the storage period, but all of them were significantly below the control (*p* < 0.05).

#### 3.1.5. Moisture Content

The moisture content of root mustard decreased continuously with the increasing storage duration, and moisture content of the 1-MCP treatment groups was higher than the control for the entire duration of storage, with 1.0 μL·L^−1^ 1-MCP having the highest moisture content (Table 1).

### 3.2. Enzyme Activity

#### 3.2.1. The Activity of PDH, LOX, PPO, and APX

PDH activity, LOX activity, and PPO activity in mustard increased during storage. Significant suppression of PDH activity, LOX activity and PPO activity were observed in 1-MCP fumigated root mustard in comparison to control (*p* < 0.05) (Figure 5A–C). On day 7 and 14, PDH and PPO activities with 1.0 μL·L^−1^ 1-MCP treatment were lower than other groups, reaching 208.9 U kg^−1^ and 50.1 U kg^−1^, respectively. Nevertheless, the LOX activity of 1.5 μL·L^−1^ 1-MCP treatment was significantly lower than that of the other two groups at the early stage of storage. Afterward, the difference between 1.0 μL·L^−1^ and 1.5 μL·L^−1^ 1-MCP treatment was insignificant, but the difference between the two groups was significantly lower than 2.0 μL·L^−1^ (*p* > 0.05). The trends in PAL activity were similar for control and treated root mustard (Figure 5D). However, the treated PAL activity values were consistently higher than the control. Among them, 1.0 μL·L^−1^ of 1-MCP fumigation was more useful in the maintenance of PAL reduction and retardation of root mustard aging.

#### 3.2.2. The Activity of SOD, SDH, POD, and APX

The activity changes of SOD, SDH, and POD showed similar trends, which increased first and then decreased. Compared to the control, 1-MCP promoted SOD and SDH activities during storage, with 1.0 μL·L^−1^ superior to the other concentrations (Figure 6A,B). The POD activity of root mustard peaked on day 14, 1.0 μL·L^−1^ had the highest activity, the peak was 61.97 U·g^−1^, next was 1.5 μL·L^−1^, the peak was 59.79 U·g^−1^, and the lowest was control group, the peak was 40.15 U·g^−1^ (Figure 6C). At the beginning of storage, APX activity of root mustard treated with 1-MCP decreased rapidly (Figure 6D). In contrast, the control group decreased slowly. Since then, APX activity remained on a downtrend in control, while it was on an upswing in the treatment groups. The APX activity of root mustard in the treated groups was obviously above that in control (*p* < 0.05). The results demonstrated that 1-MCP treatment could better maintain the reduction of APX content in root mustard at the late storage stage.

### 3.3. Correlation Analysis

#### 3.3.1. Effects of 1-MCP Treatment on Storage Quality and Enzyme Activity of Root Mustard Based on PCA Analysis

As can be seen from Figure 7, the contribution of the first principal component of root mustard PCA at different storage times in the control and 1.0 μL·L^−1^ 1-MCP-treated groups was 46.0% and the contribution of the second principal component was 31.1%, with a cumulative contribution of 77.1%, indicating that these two principal components basically reflected all the characteristics between the control and 1.0 μL·L^−1^ 1-MCP-treated root mustard sample groups. The root mustard samples of CK_14d_ and 1.0 μL·L^−1^ 1-MCP_28d_ were clustered together, indicating a high similarity between the first and second principal components of these two samples, which could laterally respond that the storage effect of 1.0 μL·L^−1^ 1-MCP-treated root mustard at day 28 of storage was similar to that of the control at 14 days of storage, indicating that 1.0 μL·L^−1^ 1-MCP could effectively prolong the storage quality of root mustard.

#### 3.3.2. Effects of 1-MCP Treatment on Storage Quality and Enzyme Activity of Root Mustard Based on OPLS-DA

The analysis of OPLS-DA in the control and 1.0 μL·L^−1^1-MCP treatment group is shown in Figure 8. The overall contribution of the changes in the indicators of the two groups to the model can be seen from the projection value (VIP). The horizontal coordinate of the S-plot indicates the covariance coefficients of the main features and metabolites, and the vertical coordinate indicates the correlation coefficients of the main components and metabolites. The farther away from the center point, the more outstanding the contribution to the differentiation of the storage effect of root mustard between the two treatments, and the greater the corresponding VIP value. Based on the VIP values greater than 1, the root mustard difference indicators could be identified as lignin, weight loss, Vit C, MDA, hardness, SOD, LOX, PDH, and POD, which indicated that these indicators were the leading landmark indicators for judging the quality differences between the control and 1.0 μL·L^−1^ 1-MCP-treated root mustard samples.

## 4. Discussion

Root mustard is a root vegetable with a high respiratory metabolism, which accelerates shriveling, yellowing of the rind, and loss of nutritional quality after harvest. In the studies indicated that 1-MCP has been extensively used in postharvest storage with its functions of inhibiting ethylene and extending storage life [44]. Currently, 1-MCP delays the ripening of several fruits, including kiwifruit [45], mangosteen (*Garcinia mangostana L.*) fruit [46], banana [47], etc. Ma et al. [48] found that 1-MCP can reduce the weight loss of Yate kiwifruit, keep high moisture and reduce rot. Based on these studies, we examined the storage quality of root mustard treated with different concentrations of 1-MCP. The results showed that different concentrations of 1-MCP treatment could inhibit the deterioration of appearance quality and weight loss of root mustard, reduced fruit rot and mildew, inhibited stem and leaf rot and abscission, delay the softening of root mustard and maintain high hardness, and 1.0 μL·L^−1^ had the best storage effectiveness. A study on apples has been conducted showing that 1-MCP treatment inhibited the hydrolysis activity of the cell wall, maintained the pectin of the apple cell wall, and decreased the dissolution of polysaccharides and neutral sugar to delay the softening process [49]. Zkaya et al. [50] also found that 1-MCP can better maintain the hardness of nectarines. In addition, we found that the epidermis of the control group turned yellow during storage; the yellowing phenomenon is more serious. In contrast, the epidermis of root mustard treated with 1-MCP was only slightly yellowed. Cheng et al. [51] found that pears treated with 1-MCP had higher chlorophyll content, complete chlorophyll, neat arrangement of grana thylakoids, and lower rate of yellow flowers in peel compared with the control. Studies also showed that six Asian vegetables treated with 1-MCP combined with exogenous ethylene significantly reduced the yellow flower rate of West Indian lime fruit, and mustard peer [52,53,54].

Respiratory metabolism and ethylene release are important physiological activities that provide nutrients and energy to sustain the normal vital processes in post-harvest fruits and vegetable harvests. They reflect the maturation and aging of fruits and vegetables during storage [55]. The role of 1-MCP in postponing ethylene manufacture and respiration rates has been broadly confirmed [56,57,58]. A lot of research has indicated that 1-MCP significantly improves post-harvest fruit hardness due to the suppression of ethylene [56,59]. Our findings are consistent with this viewpoint. Our data showed that 1.0 μL·L^−1^ 1-MCP could inhibit the respiration and ethylene release rate of root mustard, indicating that 1-MCP prevented the formation of the ethylene receptor complex and blocked the ethylene-induced signaling pathway.

In recent years, brassica vegetables have attracted much attention because of their functions as antioxidants, preventing cancer and cardiovascular diseases and delaying the aging process [60]. Liang et al. [61] found that 1-MCP treatment could maintain a low level of active oxygen in fresh jujube fruit, improve the antioxidant capacity of the fruit itself, and activate SOD, PPO and POD. Studies on fresh-cut peach and *Pyrus communis L.* showed that 1-MCP inhibited PPO activity, delayed fruit browning, and increased SOD and POD activities [62,63]. With these findings, the influence of 1-MCP on the activity of Vit C content, PPO, POD, and SOD were examined. The results suggested that 1-MCP treatment was effective in maintaining Vit C content of root mustard during storage, inhibited the PPO activity, reduced the oxidation rate of phenolic compounds, and increased the activities of POD and SOD, indicating that 1-MCP can diminish the build-up of ROS by increasing the activity of ROS metabolic enzymes and enhance the antioxidant capacity of mustard. The root mustard treated with 1.0 μL·L^−1^ 1-MCP had better quality, lower PPO activity, and better POD and SOD activity.

The changes in relative conductivity and MDA content reflected the damage degree to cell membrane. When plants are injured by stress, cell membrane lipids and membrane proteins are vulnerable to damage, resulting in a large number of intracellular substances extravasation, thereby increasing the relative conductivity [58]. MDA content is one of the primary products of membrane lipid peroxidation. The increase in MDA content indicates that membrane lipid peroxidation is enhanced and fruit senescence is aggravated [64]. LOX is a crucial enzyme causing membrane lipid peroxidation and generates ROS during degradation, and the increase in LOX activity will contribute to the accumulation of MDA and accelerate fruit senescence and ripening [65]. Xie et al. [66] found that 1-MCP treatment significantly reduced the activity of pineapple LOX and MDA levels to reduce ROS production. In this study, we found that the MDA content, Cell permeability, and LOX activity of 1-MCP-treated root mustard were lower than those of the control, showing that 1-MCP treatment slowed down the degree of cell membrane breakage of root mustard, thus slowing down the aging process of root mustard. APX plays an important role in scavenging reactive oxygen species and ascorbic acid metabolism, which can delay the aging of fruits and vegetables [67]. This function was seen many fruits such as kiwifruit [68] and pear [39] during their storage stages. SDH is a vital enzyme in the tricarboxylic acid cycle, catalyzing the dehydrogenation of succinic acid, which is absolutely required to catalyze ATP synthesis [69]. Zhou [70] also indicated that alterations in SDH activity might lead to the disorder of electron flow in the mitochondrial respiratory chain, leading to body damage. In addition, PDH is the prominent the pyruvate dehydrogenase complex, and as one of the key enzymes of aerobic respiration, the decrease in pyruvate dehydrogenase activity indicates a reduction of aerobic respiration [71]. In this experiment, the SDH activity of 1-MCP treatment was more significant than the control, and the PDH activity was lower than that of the control group. In addition, we also found that 1.0 μL·L^−1^ and 1.5 μL·L^−1^1-MCP had better effects on promoting SDH activity and inhibiting PDH activity than 2.0 μL·L^−1^1-MCP treatment. The results showed that as an important enzyme involved in energy metabolism, the increase in SDH activity and the decrease in PDH activity might help alleviate the senescence of mustard through 1-MCP treatment.

## 5. Conclusions

In this experiment, 1.0 μL·L^−1^,1.5 μL·L^−1^, and 2.0 μL·L^−1^ 1-MCP were used to treat root mustard, and the changes in physiological indicators and associated enzyme activities were analyzed over the storage period. The results indicated that, compared to the control, the three groups of 1-MCP could effectively slow down the decrease in weight loss rate, hardness, lignin, and Vit C content, inhibit respiration intensity, reduce ethylene release, decrease cell permeability and MDA accumulation, maintain cell membrane integrity, inhibit the increase in LOX, PDH, and PPO activities, and maintain SDH, SOD, APX, PAL, and POD activities at a high level. The effect of 1.0 μL·L^−1^1-MCP was better than 1.5 μL·L^−1^ and 2.0μL·L^−1^1-MCP. In summary, 1.0 μL·L^−1^1-MCP treatment delayed the senescence and quality deterioration of mustard and maintained good edible quality and nutritional value after 35 days of storage. At the same time, 1-MCP is a non-toxic, trace amount of a safe chemical preservative that is easy to handle and economical to use.

## Figures and Tables

**Figure 1 foods-11-02978-f001:**
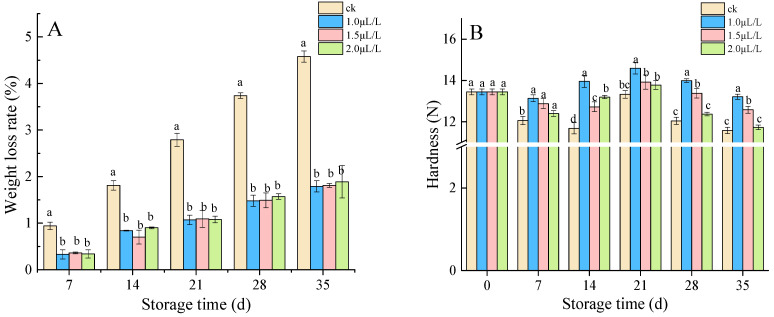
Effects of 1-MCP on weight loss (**A**) and hardness (**B**) of root mustard. Different letters at each time point indicated a significant difference between treatments (*p* < 0.05) by Duncan’s multiple range test.

**Figure 2 foods-11-02978-f002:**
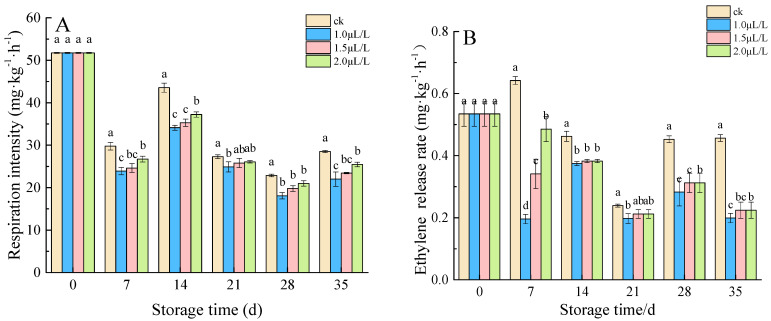
Effects of 1-MCP on respiratory intensity (**A**) and ethylene content (**B**) in root mustard. Different letters at each time point indicated a significant difference between treatments (*p* < 0.05) by Duncan’s multiple range test.

**Figure 3 foods-11-02978-f003:**
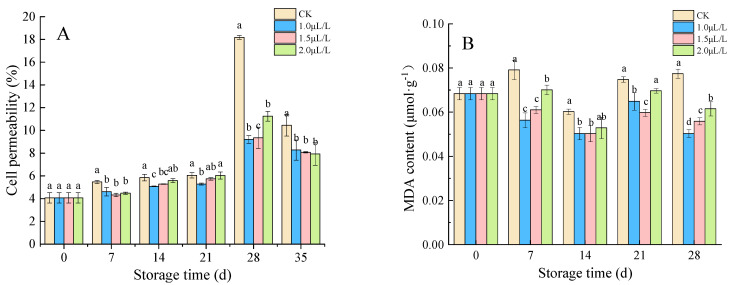
Effects of 1-MCP on cell permeability (**A**) and MDA content (**B**) in root mustard. Different letters at each time point indicated a significant difference between treatments (*p* < 0.05) by Duncan’s multiple range test.

**Figure 4 foods-11-02978-f004:**
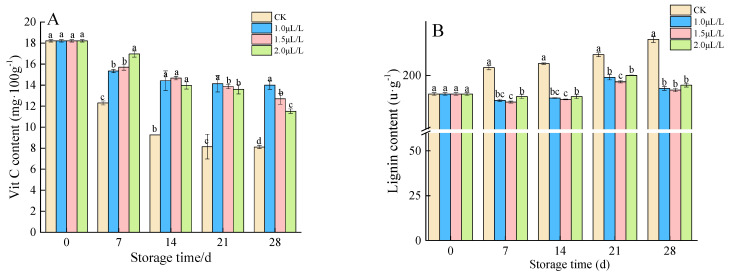
Effects of 1-MCP on Vit C content (**A**) and lignin content (**B**) of root mustard. Different letters at each time point indicated a significant difference between treatments (*p* < 0.05) by Duncan’s multiple range test.

**Figure 5 foods-11-02978-f005:**
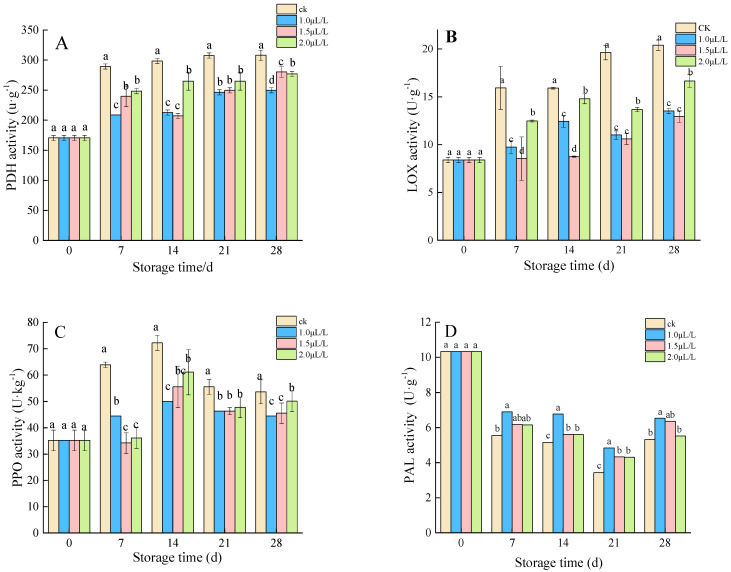
Effects of 1-MCP on PDH activity (**A**), LOX activity (**B**), PPO activity (**C**), and PAL activity (**D**). Different letters at each time point indicated a significant difference between treatments (*p* < 0.05) by Duncan’s multiple range test.

**Figure 6 foods-11-02978-f006:**
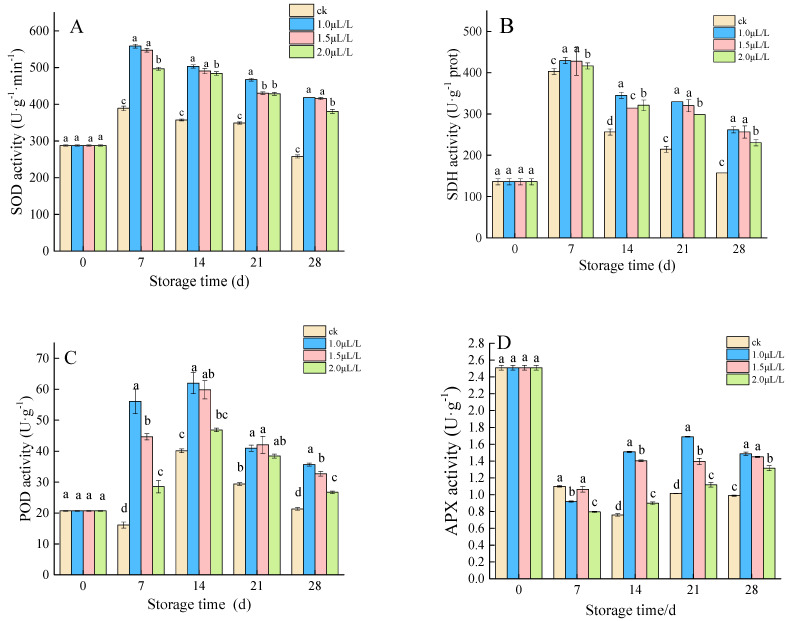
Effects of 1-MCP on SOD activity (**A**), SDH activity (**B**), POD activity (**C**) and APX activity (**D**). Different letters at each time point indicated a significant difference between treatments (*p* < 0.05) by Duncan’s multiple range test.

**Figure 7 foods-11-02978-f007:**
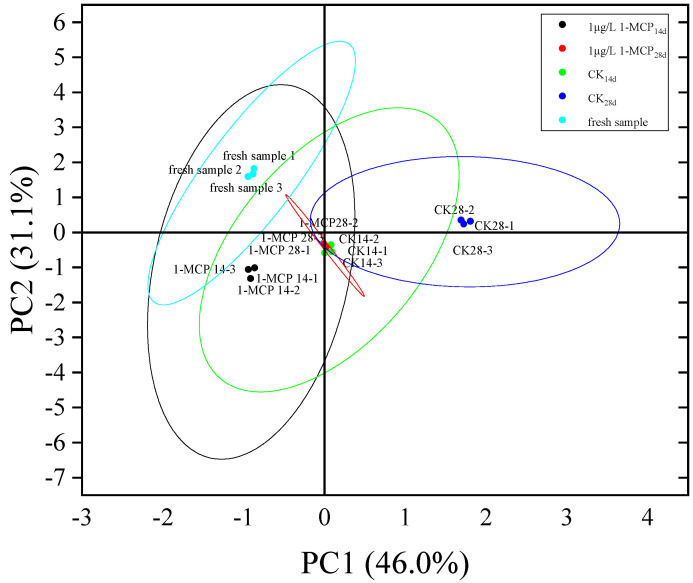
PCA analysis of storage quality and enzyme activity of root mustard at different storage times for control and 1.0 μL·L^−1^ 1-MCP. Ellipse represent confidence intervals.

**Figure 8 foods-11-02978-f008:**
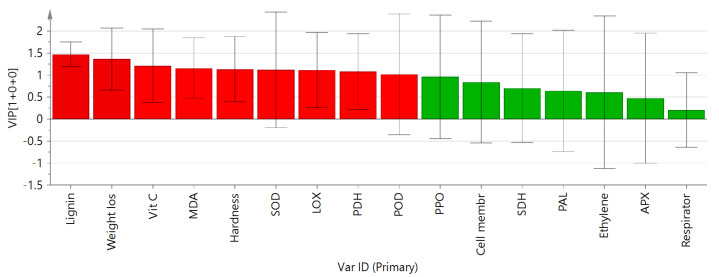
VIP and S-plot plots of storage quality and enzyme activity of root mustard treated with control and 1.0 μL·L^−1^ 1-MCP during ambient temperature storage. Red represents VIP values greater than 1, and green represents VIP values greater than 1.

**Table 1 foods-11-02978-t001:** Effect of 1-MCP on moisture content of root mustard.

	Treatment	CK	1.0 μL·L^−1^	1.5 μL·L^−1^	2 μL·L^−1^
Time/d					
0		91.22 ± 0.12 ^a^	91.22 ± 0.12 ^a^	91.22 ± 0.12 ^a^	91.22 ± 0.12 ^a^
7		88.88 ± 0.04 ^d^	90.51 ± 0.04 ^b^	90.38 ± 0.03 ^c^	90.99 ± 0.07 ^a^
14		88.16 ± 0.05 ^b^	89.28 ± 0.06 ^a^	89.25 ± 0.09 ^a^	89.25 ± 0.08 ^a^
21		87.39 ± 0.17 ^c^	89.76 ± 0.13 ^a^	89.01 ± 0.06 ^b^	89.08 ± 0.07 ^b^
28		86.82 ± 0.31 ^c^	89.61 ± 0.18 ^a^	88.84 ± 0.20 ^b^	88.76 ± 0.08 ^b^
35		86.24 ± 0.10 ^d^	89.48 ± 0.12 ^a^	88.69 ± 0.12 ^b^	88.46 ± 0.06 ^c^

Different letters at each time point indicated a significant difference between treatments (*p* < 0.05) by Duncan’s multiple range test.

## Data Availability

Data is contained within the article.

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
