# Peer review of "Effects of 1-MCP Treatment on Physiology and Storage Quality of Root Mustard at Ambient Temperature"

_foods, 2022, doi:10.3390/foods11192978_

Round 1

Reviewer 1 Report

The idea of the manuscript is so good and very important to the postharvest sector. the strength of it belongs to the application of 1-MCP on Mustard root only because there is a lot of paper discussing the effect of 1-MCP as the postharvest treatment on other fruit and vegetables.   anyway, I can't recommend this manuscript on the current status. It must be improved:

1. you should define the abbreviation used in the abstract such as 1-MCP "1-Methycyclopropene."

2. line 11-12, the storage conditions were defined without the storage period! also, in the method part (sample handling)

3. The introduction is short and missing the previous technology with details used for root mustard preservation and its effect on the mustard quality

4. Also, I think it will be better if you add more details about the previous studies that used 1-MCP as a postharvest treatment in the introduction part and discussion.

5. almost all methods need to rewrite in detail to be easily repeated by other researchers 

6. there is no figure or table caption for the statistical analysis. Moreover, the results will be clear and helpful if you make statistical analysis between treatments and during storage to show the combined effect between them 

7. discussion was soo poor because it  is missing citations from previous studies, and it needs to add more explanation

8.  study  should mention if the usage of 1-MCP was economical or not  in the conclusion part  

Reviewer 2 Report

This study could serve as a theoretical reference for root mustard preservation with

1-     MCP treatment.

1.      In addition to line 260-262, Changes in appearance or smell, sensory test results could be described.

2.      Rationale of 1-MCP concentration to be given and discussed in detail by comparing previous study of other root vegetables. Looks not so much big difference in applied concentration range, but what will happen if lower dosage of 0.1 or 0.5?

3.      Explain why ethylene content can be measured by carbon dioxide analyzer? (Line 91-93)

4.      Fig.3 a, why cell permeability significantly increased at day28?

5.      Fig.10 should be Fig.7. By the way, it would be kind to explain relationship of VIP and S-plot after primary component analysis. What is the difference in color of green and red?

Reviewer 3 Report

This research is interested. Application of 1-MCP is important for many fruit and vegetable storage if you find the best dose and standardized the treatment. The work is well-written and clear results have been described. I have some minor comments as follow:

Line 12 : how long was the treatment lasts? And Root mustard has been stored for how long?

Line 28 what country? Write, China

Line 33 remove ‘by’

Line 37 need citation. 

Line 41 remove ‘novel’ 1-MCP is known since 2006.

LINE 55 is it cold storage? Need citation and extend the paragraph by explaining more details of these research at room temperature as this directly related to your crop.

Line 67 give more detail about the 1-MCP formulation and preparation before use

Line 68 is it perforated plastic bag?

Line 92 Ethylene is measure by carbon dioxide analyzer?

Line 97 briefly describe the method

Line 98 better use Vit C not Vc

Line 116 citation is needed and brief description of the method

Line 145 what is ck? Is it the control? For the statistical analysis is it one way ANOVA between treatments within a time point of storage, or two ways difference between the storage time points also analyzed?.

Line 325 the best treatment you should mention for how long.

Reviewer 4 Report

In the sentence: with various concentrations of 1-MCP (1 µL 11 1, 1.5 µL L-1 , and 2.0 µL L-1 , and the text). Another option is to write with various concentrations of the sample 1-MCP, (expressed in µL L-1), and they were 1, 1.5 and 2 respectively.

The use of flowcharts is recommended for some of the main procedures.

It is very important to mention, if there is any standard rule, to carry out the measurement, or is it an independent determination without standardizing, related to this phrase: The central part of the root 78 mustard (1 cm × 2 cm × 1 cm) was taken and placed on a plate,

Take care of the spacing in the references specifically in line 80. of 100 mm min-125, 26, 9 80

Add a space to the value and the degree symbol, t 17°C, on line 84.

In line 88 m represents sample mas, in this sentence check if you want to clarify that it is a unit of mass.

Take care of the format of the use of references both in the text and in the exclusive part of the references, for example in the line Wei.[27]., line 95.

Write the meanings of the acronyms when they first appear in the text of the writing, for readers from the same area, this is common, but for those who are not, this represents breaking the sequence of reading. For example, in the line PPO and POD activities were measured by Cao.[31] Lines 122 and 123.

Revisar esta tendencia: which 1.0 μL L-1 was the lowest (Fig. 2A)

Please, that the letters in the legends of the graphs are the same as those of the axes

Specify the meaning of Vc, The content of Vc ethylene and extending storage life[32]

On line 255, write in parentheses the short meaning of the term “of rhizome vegetables”,

.

It is also recommended to use photographs or photomicrographs, at least. in the initial stages, and the ideal would be, Also, from the samples with the best results obtained, it would be much more illustrative.

In line 70 revisar the term "text"

Round 2

Reviewer 1 Report

Thanks for your great efforts. The authors have completed all required modifications.

Reviewer 3 Report

Thank you. the author replied to the comment satisfactory.